# Treatment of Navicular Stress Fracture Accompanied by Os Supranaviculare: A Case Report

**DOI:** 10.3390/medicina58010027

**Published:** 2021-12-24

**Authors:** Woo-Jong Kim, Ki-Jin Jung, Eui-Dong Yeo, Hong-Seop Lee, Sung-Hun Won, Dhong-Won Lee, Jae-Young Ji, Sung-Joon Yoon, Yong-Cheol Hong

**Affiliations:** 1Department of Orthopaedic Surgery, Soonchunhyang University Hospital Cheonan, Cheonan 31151, Korea; kwj9383@hanmail.net (W.-J.K.); c89546@schmc.ac.kr (K.-J.J.); yunsj0103@naver.com (S.-J.Y.); 2Veterans Health Service Medical Center, Department of Orthopaedic Surgery, Seoul 05368, Korea; angel_doctor@naver.com; 3Nowon Eulji Medical Center, Department of Foot and Ankle Surgery, Eulji University, Seoul 01830, Korea; sup4036@naver.com; 4Department of Orthopaedic Surgery, Soonchunhyang University Hospital Seoul, Seoul 04401, Korea; orthowon@schmc.ac.kr; 5Department of Orthopaedic Surgery, Konkuk University Medical Center, Seoul 05030, Korea; bestal@naver.com; 6Department of Anesthesiology and Pain Medicine, Soonchunhyang University Hospital Cheonan, Cheonan 31151, Korea; phmjjy@naver.com

**Keywords:** os supranaviculare, tarsal navicular, stress fracture

## Abstract

Navicular stress fractures (NSFs) are relatively uncommon, and predominantly affect athletes. Patients complain of vague pain, bruising, and swelling in the dorsal aspect of the midfoot. Os supranaviculare (OSSN) is an accessory ossicle located above the dorsal aspect of the talonavicular joint. There have been few previous reports of NSFs accompanied by OSSN. Herein we report the case of a patient with OSSN who was successfully treated for an NSF. A 34-year-old Asian man presented with a 6-month history of insidious-onset dorsal foot pain that occasionally radiated medially toward the arch. The pain worsened while sprinting and kicking a soccer ball with the instep, whereas it was temporarily relieved by rest for a week and analgesics. Plain radiographs of the weight-bearing foot and ankle joints revealed a bilateral, well-corticated OSSN. Computed tomography (CT) revealed a sagittally oriented incomplete fracture that extended from the dorsoproximal cortex to the center of the body of the navicular. The OSSN was excised and the joint was immobilized with a non-weight-bearing cast for 6 weeks, followed by gradual weight bearing using a boot. The 5-month follow-up CT scan demonstrated definite fracture healing. At the 1-year follow-up, the patient’s symptoms had resolved, the American Orthopedic Foot and Ankle Society midfoot score had improved from 61 to 95 points, and the visual analog scale pain score had improved from 6 to 0. We describe a rare case of NSF accompanied by OSSN. Because of the fracture gap and biomechanical properties of OSSN, OSSN was excised and the joint was immobilized, leading to a successful outcome. Further research is required to evaluate the relationship between NSFs and OSSN, and determine the optimal management of NSFs in patients with OSSN.

## 1. Introduction

Navicular stress fracture (NSF) was first described by Towne et al. in 1970 [1]. The fracture lines are typically oriented sagittally and originate from the dorsal proximal cortex of the navicular bone. NSFs are relatively rare, and predominantly affect athletes involved in vigorous jumping, running, and sprinting [2]. Patients complain of vague pain, bruising, and swelling in the dorsal aspect of the midfoot. The pain is aggravated by running and relieved by rest.

Os supranaviculare (OSSN), also called Pirie’s bone, is an accessory ossicle located above the dorsal aspect of the talonavicular joint. It is derived from an accessory ossification center, and is found in 1% of the population [3]. Although many studies on the topic of NSF have been published, reports of NSF accompanying OSSN are extremely rare [2,4,5]. Ingalls and his colleagues first published a paper on the radiological diagnosis of NSF with OSSN in 2011. They diagnosed using X-ray, CT, MRI, and bone scan, but only inferred the correlation between NSF and OSSN without mentioning conservative treatment or surgery [4]. Kiter et al. reported that the symptoms disappeared after OSSN removal in a patient who visited the hospital for dorsal foot pain, but there was no NSF in this case [5]. In addition, treatment and bone union outcomes of NSF associated with OSSN have not been reported. Thus, here we report a patient with NSF accompanied by OSSN who achieved bone union through surgical resection.

## 2. Case Presentation

This case report was approved by the Institutional Review Board of Soonchunhyang University Hospital, South Korea (no. 2021-06-008). The patient provided written informed consent for the publication of this report and the accompanying images.

A 34-year-old Asian man presented with a 6-month history of insidious-onset dorsal foot pain that occasionally radiated medially toward the arch. The patient was a left-footed amateur soccer player, who had participated in 1–2 full-time soccer matches, as well as 2–3-h practice sessions, every week for 7 years. The patient’s pain worsened while sprinting and kicking a soccer ball with the instep. The pain was temporarily relieved with rest for a week and use of analgesics. He was otherwise healthy, with no history of foot trauma or any other diseases.

Physical examination revealed focal tenderness over the dorsal aspect of the talonavicular joint, and minimal limitation in the range of motion of the foot and ankle. There was mild swelling, but no other signs of inflammation. The pain was exacerbated by forced ankle dorsiflexion, forefoot inversion, the single-leg hop test, and the toe standing test. The American Orthopedic Foot and Ankle Society (AOFAS) midfoot score (=69) and visual analog scale (VAS) pain score (=6) were obtained. There were no symptoms related to the contralateral foot.

Plain radiographs of the weight-bearing foot and ankle joint revealed well-corticated bilateral OSSN above the proximal aspect of the navicular bone, but no definite fracture line or other abnormality (Figure 1). A three-dimensional computed tomography (CT) scan revealed well-corticated ossicles in the osseous depression of the dorsal proximal margin (Figure 2a). Multiple bony cysts were observed under the sclerotic margin surrounding the osseous depression in the left foot (Figure 2b). The sagittal image of the left foot revealed an incomplete fracture line, oriented sagittally and extending from the dorsoproximal cortex to the center of the navicular body (Figure 2c). Magnetic resonance imaging (MRI) confirmed the fracture, with high signal intensity at the fracture location (Figure 3). Considering the chronicity of the patient’s symptoms, as well as the surrounding sclerosis and bony cysts observed on the CT scan, the patient was diagnosed with NSF and OSSN.

The patient was managed non-surgically. A mixture of 5 mg lidocaine and 0.5 mg triamcinolone acetate was injected around the OSSN under ultrasound guidance. The pain was immediately relieved, but recurred after 24 h. As the next step, non-weight bearing below-knee cast immobilization was performed for 6 weeks, but the patient’s symptoms did not improve. Because of the failure of non-surgical treatment, the OSSN was excised to eliminate the impingement of OSSN on the talonavicular joint and reduce the possibility of NSF nonunion. Under general anesthesia, a dorsal linear skin incision was made between tendons of the extensor hallucis brevis (EHB) and the extensor hallucis longus. The neurovascular bundle containing the deep peroneal nerve and dorsalis pedis artery and vein was retracted, and the talonavicular joint capsule and OSSN were excised (Figure 4). The synovium around the ossicle appeared thick and irritated. The synovium was debrided, as it was considered a possible source of pain.

Postoperatively, the foot and ankle joints were immobilized in a non-weight-bearing below-knee cast for 6 weeks, followed by gradual weight bearing for 2 weeks, using boots that allowed full range of motion. At the 4-month follow-up, the patient did not have any discomfort, and could participate in sports. The CT scan performed at the 5-month follow-up revealed definite fracture healing (Figure 5). At the 1-year follow-up, the patient’s symptoms had resolved, and the AOFAS midfoot score (61 to 95 points) and VAS score (6 to 0) had improved.

## 3. Discussion

The tarsal navicular is a saddle- or boat-shaped bone located between the three cuneiforms and the talar head. It has a unique oblique axis, with the base situated dorsolaterally and apex oriented plantarmedially [6]. The most prominent part of this bone is a medial tuberosity, which serves as an attachment for the tibialis posterior tendon and plantar navicular ligaments. Laterally, its small and convex surface articulates with the cuboid, while its proximal and distal surfaces articulate with the talar head and three cuneiforms, respectively. The spring (plantar calcaneonavicular) ligament inserts into the plantar side of the navicular, and the bifurcate ligaments attach to its superolateral surface.

The navicular receives dual blood supply from the tibialis posterior and dorsalis pedis arteries. Blood is supplied to the dorsal surface by the medial tarsal branch of the dorsalis pedis artery, and to the medial plantar aspect by a branch of the posterior tibial artery. The central one-third of the bone consists of a hypovascular zone, which predisposes the tarsal navicular to stress fractures, delayed healing, and nonunion [7,8]. The unique anatomical location of the navicular also predisposes it to stress fractures. During foot strike, medial compression forces transmitted from the second metatarsocuneiform joint are partially transmitted to the talar head, whereas lateral compressive forces from the first metatarsocuneiform joint are transmitted to the navicular bone alone. This uneven force distribution concentrates the shear stress on the central region of the bone [7,9]. Therefore, the limited vascularity and unique biomechanics explain the vulnerability of the central region to stress fractures.

Early research reported that NSF accounts for 0.7–2.4% of lower extremity stress fractures. However, recent studies reported that the incidence has increased up to 35% due to increased awareness of NSF and improved diagnostic modalities [10,11]. Several factors increase the risk for NSFs. Pavlov et al. reported that patients with NSF were more likely to have metatarsus adductus, short first metatarsal, or a relatively long second metatarsal [12]. Hossain et al. reported that patients with reduced ankle dorsiflexion have a higher risk for NSF because of the compensatory increase in navicular motion leading to greater shear stress on the central third of the navicular. The os supranaviculare (OSSN), also called Pirie’s bone is an accessory bone that occurs in about 1% of the population [3]. It is usually located close to the middle of the talonavicular joint where compressive forces are shared with the talar head and the navicular bone [13]. Since OSSN itself can cause pain in the dorsum of the foot, proper diagnosis and treatment are required even if it is not accompanied by NSF [5]. Because the OSSN is located between the talar head and navicular, it displaces the central part of the navicular distally during foot strike, resulting in greater shear stress being transmitted to the central third of the navicular. The OSSN may increase the stress on the navicular under normal loading conditions (e.g., while kicking a ball). Although our patient had bilateral OSSN, NSF occurred only in the left foot (which was mainly used to kick the ball), which suggests that OSSN increases the risk for NSF.

NSF has non-specific clinical features that make it difficult to diagnose and often lead to a delay in diagnosis of more than 6 months [2,14]. The most common symptoms are an insidious-onset sharp pain, cramps, or soreness in the anterior ankle or dorsomedial midfoot, exacerbated by jumping or sprinting and relieved by rest [15,16]. Many patients instinctively alter their gait to reduce foot strike, which minimizes the pain. Therefore, patients may initially experience pain only during sports, making early diagnosis of NSF difficult [17]. Patients may have maximal tenderness at a nickel-sized area over the prominence of the dorsal talonavicular joint, called the “N-spot” [18]. Pain may worsen during forefoot eversion or inversion and talar adduction or abduction. However, the range of ankle motion is usually normal [19].

Plain weight-bearing radiographs are usually performed first to evaluate fractures and other abnormalities. However, they should not be used alone because of the high false-negative rate of 82% [20]. Fractures may not be clearly visible on plain radiographs until significant bone resorption has occurred [21]. Bone scans have high sensitivity (up to 100%), but low specificity, for identifying NSFs. Additionally, bone scans do not accurately assess the fracture pattern [22]. MRI has high sensitivity for the detection of stress fractures. Moreover, MRI helps in the differential diagnosis with other conditions such as osteonecrosis, infections and Mueller–Weiss disease, with high sensibility to detect intraosseous edema and early perinavicular arthritic changes [23]. It also has the advantage of involving no radiation exposure. CT is regarded as the gold standard modality because of its high resolution for fracture lines, and accurate visualization of the surrounding sclerosis and cysts [18,24]. CT scans can also be used to monitor patients for fracture union and nonunion.

The treatment of NSF is usually challenging because persistent pain, nonunion, and delayed union are common [25]. There is no consensus on the ideal treatment for NSF (nonsurgical or surgical). Nonsurgical treatment options include non-weight bearing below-knee cast immobilization for 6 weeks, followed by use of a boot to allow gradual weight-bearing for 2–6 weeks [26]. Some researchers have reported high success rates of conservative treatment, regardless of fracture displacement or severity [11,20,26]. In contrast, Saxena et al. reported that the time required for healing depends on the fracture severity. Therefore, conservative treatment is appropriate if the fracture line is limited to the cortex, but surgery is required if the fracture line extends to the center of the body or the opposite cortex [25]. Surgical treatment for NSF involves open reduction and internal fixation using a solid 4.0-mm partially threaded, cannulated screw, followed by postoperative non-weight bearing casting [19]. Importantly, there is no consensus regarding the treatment for NSF in patients with OSSN [4].

If there is a sharp surface or discontinuity in the cortical bone, it can act as a stress riser and cause peripheral pain or fracture even if it is a normal morphological feature [27]. In the presence of OSSN, the dorsal surface of the navicular is depressed, and OSSN acts as a stress riser during foot sprinting and kicking, which may cause a stress fracture of the navicular bone. In this case, as a conservative treatment, non-weight-bearing below-knee cast immobilization was performed for 6 weeks, but failed. The authors believe that this is because the unremoved OSSN continuously acted as a stress riser and interfered with bone union. Therefore, we decided to surgically remove the OSSN.

Because the fracture in the aforementioned patient was incomplete and nondisplaced, we excised the OSSN without navicular reduction or fixation. Additionally, OSSN increases the stress and shear force transmitted to the central third of navicular. Therefore, removal of the OSSN facilitates the healing of NSF. During ankle motion, OSSN impinges between the navicular and talar head, and may be an additional source of the pain. Therefore, the removal of the OSSN was expected to reduce the pain in this case. The strength of this report is that successful treatment (bone union and return to sports) for NSF is described for a patient with OSSN, which is a rare condition.

## 4. Conclusions

We describe a rare case of NSF in a patient with OSSN. Because of the fracture gap and biomechanical properties of OSSN, the OSSN was excised, followed by immobilization in a non-weight-bearing cast. The patient achieved bone union and return to activity without persistent pain.

Studies with a large sample size are required to evaluate the relationship between OSSN and NSF, and to determine the optimal treatment of NSF in patients with OSSN.

## Figures and Tables

**Figure 1 medicina-58-00027-f001:**
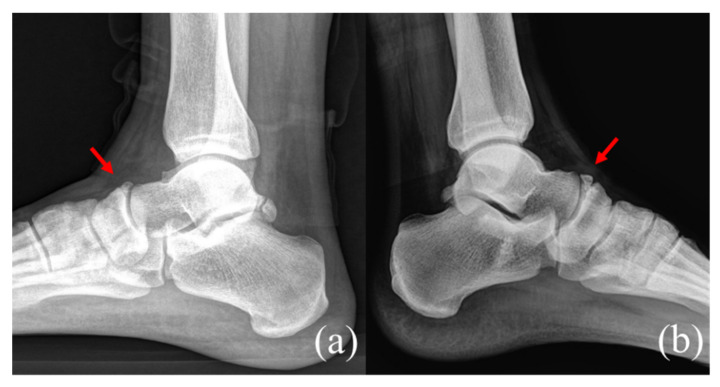
Plain lateral radiographs of the bilateral feet show the presence of well-corticated os supranaviculare (arrow) at dorsal aspect of the talonavicular without a definite fracture line. (**a**) the right and (**b**) left foot.

**Figure 2 medicina-58-00027-f002:**
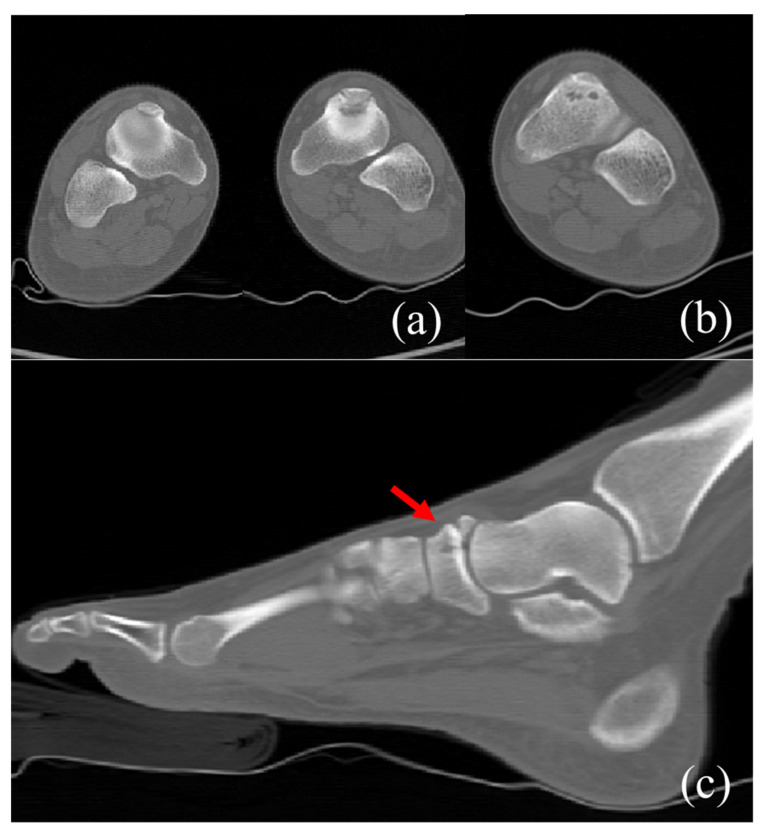
Preoperative coronal CT image (**a**) shows bilateral os supranaviculare situated in osseous depression of dorsal cortex. In the left foot, multiple bony cysts were observed under the cortex of navicular bone in the bilateral feet. In the left foot, multiple bony cysts under sclerotic margin were noted. (**b**) Preoperative sagittal CT image of the left foot (**c**) demonstrates an incomplete fracture line (arrow) extending from dorsoproximal cortex to the center of a body.

**Figure 3 medicina-58-00027-f003:**
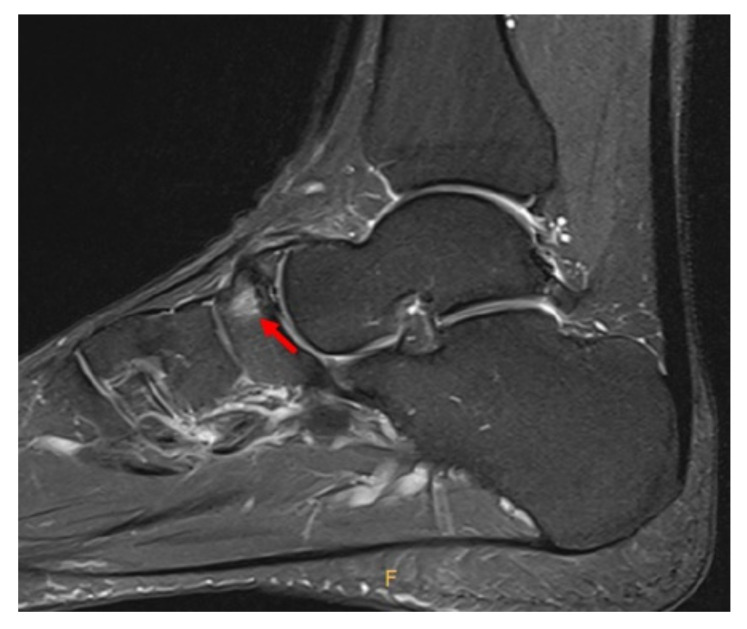
Preoperative proton density fat saturation MRI shows the fracture with high signal intensity at the same location as seen on CT scan (arrow).

**Figure 4 medicina-58-00027-f004:**
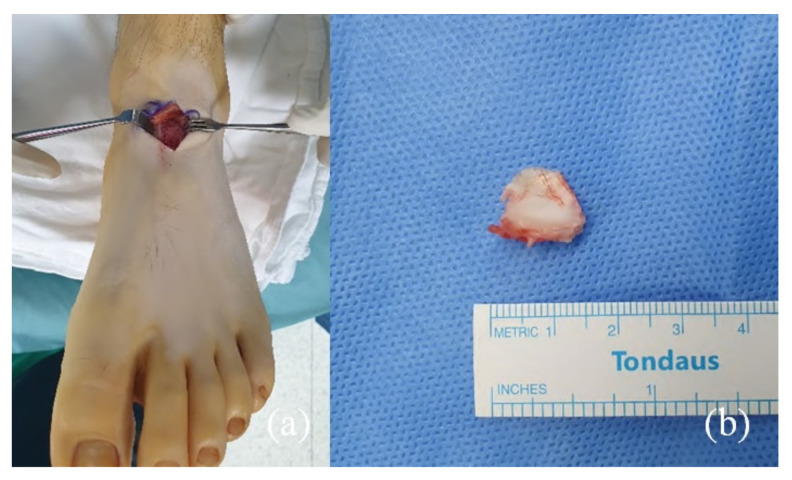
A linear incision was made through an interval between the extensor hallucis brevis (EHB) and the extensor hallucis longus (EHL) tendons (**a**). The diameter of the removed OSSN was measured to be about 1.5 cm (**b**).

**Figure 5 medicina-58-00027-f005:**
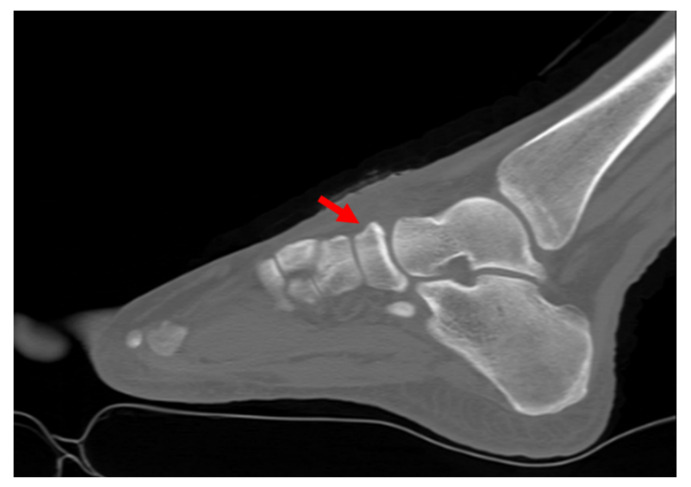
Postoperative sagittal CT image at the 5-month follow-up examination shows that bony union has been achieved (arrow).

## Data Availability

Data sharing is not applicable to this article as no datasets were generated or analyzed during the current study.

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
