# Peer review of "Treatment of Navicular Stress Fracture Accompanied by Os Supranaviculare: A Case Report"

_medicina, 2021, doi:10.3390/medicina58010027_

Round 1

Reviewer 1 Report

Treatment of navicular stress fracture accompanied by os supranaviculare: A case report.

The manuscript focuses on a rare and still rather unrecognized pathology. The methodology is correct and well written in English.

The conclusions are in line with the reported case report.

Overall, it is worthy of publication in the journal after minor revision with a couple of corrections:

- Caption figure 2 (correct "ultiple" with "multiple").

- In the section of instrumental diagnosis (lines 174-185) add that MRI helps in the differential diagnosis with other conditions such as osteonecrosis, infections and Mueller-Weiss disease, with high sensibility to detect intraosseous edema and early perinavicular arthritic changes

(Perisano C, Greco T, Vitiello R, Maccauro G, Liuzza F, Tamburelli FC, Forconi F. Mueller-Weiss disease: review of the literature. J Biol Regul Homeost Agents. 2018 Nov-Dec;32(6 Suppl. 1):157-162. PMID: 30644297).

Author Response

Response to Reviewer 1 Comments

First of all, the authors would like to thank the reviewer for the excellent advices.

Point 1: Caption figure 2 (correct "ultiple" with "multiple").

Response 1: Thanks for the good point. We corrected the spelling of the word.

Point 2: In the section of instrumental diagnosis (lines 174-185) add that MRI helps in the differential diagnosis with other conditions such as osteonecrosis, infections and Mueller-Weiss disease, with high sensibility to detect intraosseous edema and early perinavicular arthritic changes

Response 2: Thanks to you for pointing out a good paper, I was able to edit the content. We have added what the reviewer said to the text.

(Perisano C, Greco T, Vitiello R, Maccauro G, Liuzza F, Tamburelli FC, Forconi F. Mueller-Weiss disease: review of the literature. J Biol Regul Homeost Agents. 2018 Nov-Dec;32(6 Suppl. 1):157-162. PMID: 30644297).

Reviewer 2 Report

Interesting article on a little-described topic. I have a few comments: Introduction: please expand, please add more publications describing the biomechanics and pathology of the supranaviculare os in foot. Please describe in more detail what the previous articles about os Navicular stress fracture with os supranaviculare presented (items: 2,4,5). Discussion: Please describe in more detail what the previous articles about os Navicular stress fracture with os supranaviculare presented (items: 2,4,5). Please describe in detail the reasons for the possible failure of your patient's conservative treatment and why he required surgery. How os supranaviculare negatively affected the healing of Navicular stress fracture.

Author Response

Response to Reviewer 2 Comments

First of all, the authors would like to thank the reviewer for the excellent advices.

Point 1: Introduction: please expand, please add more publications describing the biomechanics and pathology of the supranaviculare os in foot. Please describe in more detail what the previous articles about os Navicular stress fracture with os supranaviculare presented (items: 2,4,5).

Response 1: Thanks for the good point. We fully agree with the point that the content of the introduction was insufficient. As you pointed out, we have added a description of the previously published paper on navicular stress fracture with os supranaviculare.

Point 2: Discussion: Please describe in more detail what the previous articles about os Navicular stress fracture with os supranaviculare presented (items: 2,4,5). Please describe in detail the reasons for the possible failure of your patient's conservative treatment and why he required surgery. How os supranaviculare negatively affected the healing of Navicular stress fracture.

Response 2: Thanks for the good advice. As you pointed out, the reason for the failure of conservative treatment was to be explained by adding a new reference(28), which may represent that the authors had no choice but to perform surgery. Also, we have added a description of the previously published paper on navicular stress fracture with os supranaviculare.

Round 2

Reviewer 2 Report

The authors have incorporated all suggested changes. Acceptable manuscript as is.